# Synthesis and Identification of New *N*,*N*-Disubstituted Thiourea, and Thiazolidinone Scaffolds Based on Quinolone Moiety as Urease Inhibitor

**DOI:** 10.3390/molecules27207126

**Published:** 2022-10-21

**Authors:** Yaseen A. M. M. Elshaier, Ashraf A. Aly, Mohamed Abdel-Aziz, Hazem M. Fathy, Alan B. Brown, Stefan Bräse, Mohamed Ramadan

**Affiliations:** 1Organic and Medicinal Chemistry Department, Faculty of Pharmacy, University of Sadat City, Menoufia 32958, Egypt; 2Department of Chemistry, Faculty of Science, Minia University, El-Minia 61519, Egypt; 3Medicinal Chemistry Department, Faculty of Pharmacy, Minia University, El-Minia 61519, Egypt; 4Pharmaceutical Organic Chemistry Department, Faculty of Pharmacy, Al-Azhar University, Assiut Branch, Assuit 71524, Egypt; 5Chemistry Department, Florida Institute of Technology, Melbourne, FL 32901, USA; 6Institute of Organic Chemistry, Karlsruhe Institute of Technology, 76131 Karlsruhe, Germany; 7Institute of Biological and Chemical Systems (IBCS-FMS), Karlsruhe Institute of Technology, 76344 Eggenstein Leopoldshafen, Germany

**Keywords:** *N*,*N*-disubstituted thioureas, dialkyl acetylenedicarboxylates, thiazolidinones, in vitro urease inhibition properties, molecular docking

## Abstract

Synthesis of thiazolidinone based on quinolone moiety was established starting from 4-hydroxyquinol-2-ones. The strategy started with the reaction of ethyl bromoacetate with 4-hydroxyquinoline to give the corresponding ethyl oxoquinolinyl acetates, which reacted with hydrazine hydrate to afford the hydrazide derivatives. Subsequently, hydrazides reacted with isothiocyanate derivatives to give the corresponding *N*,*N*-disubstituted thioureas. Finally, on subjecting the *N*,*N*-disubstituted thioureas with dialkyl acetylenedicarboxylates, cyclization occurred, and thiazolidinone derivatives were obtained in good yields. The two series based on quinolone moiety, one containing *N*,*N*-disubstituted thioureas and the other containing thiazolidinone functionalities, were screened for their in vitro urease inhibition properties using thiourea and acetohydroxamic acid as standard inhibitors. The inhibition values of the synthesized thioureas and thiazolidinones exhibited moderate to good inhibitory effects. The structure−activity relationship revealed that *N*-methyl quinolonyl moiety exhibited a superior effect, since it was proved to be the most potent inhibitor in the present series achieving (IC_50_ = 1.83 ± 0.79 µM). The previous compound exhibited relatively much greater activity, being approximately 12-fold more potent than thiourea and acetohydroxamic acid as references. Molecular docking analysis showed a good protein−ligand interaction profile against the urease target (PDBID: 4UBP), emphasizing the electronic and geometric effect of *N*,*N*-disubstituted thiourea.

## 1. Introduction

Urease is a well-recognized enzyme that hydrolyzes urea to ammonia and carbon dioxide in living organisms. It is found in fungi, bacteria, plants, and vertebrates [1]. The amount of ammonia generated during hydrolysis tends to raise the pH, and an increase in medium pH is linked to the development of a variety of health problems in people who depend on colonization sites by urease-producing microorganisms [2,3]. It is known that the ureolytic activity of several microorganisms, e.g., *Proteus mirabilis*, is related to the formation of urinary tract stones, which can lead to chronic kidney and pelvic inflammation. In addition, urinary catheter obstruction in patients results in the colonization of urease-producing microorganisms, primarily *P. mirabilis*. Infectious bacteria that produce too much ammonia can cause ammonia encephalopathy or hepatic coma [4,5]. Another mechanism by which urease participates in pathogenic bacteria infection is the establishment of a microenvironment favorable to pathogen survival [6].

Due to the role of urease in such clinically significant complications, urease activity must be regulated using inhibitors [1,7]. Several classes of compounds have been identified as urease inhibitors [8]. Because of the association of ureases with several pathological conditions [9], the discovery of effective and safe urease inhibitors has been an important area of pharmaceutical research. 

Thiosemicarbazide derivatives have been prominent precursors for synthesizing nitrogen and sulfur-containing heterocyclic compounds in recent decades due to their abundance of reactive centers [10]. In addition, thiosemicarbazones have also been evaluated as ribonucleotide reductase inhibitors and exhibit potential as anticancer drugs similar to methisazone and triapine [11,12,13,14,15]. These sulfur and nitrogen donor ligands and their coordination complexes have attracted significant attention due to their activity against the smallpox virus and protozoa influenza [16]. Many studies have recently been published on the efficacy of thiosemicarbazides and their hybrid derivatives in suppressing urease enzyme activity [17,18].

The current study is a part of our ongoing research into the synthesis of bioactive hybrid molecules, and a continuation of our previous work on the design of antibacterial urease inhibitors [19]. That previous work was involved the synthesis of 3-thiosemicarbazides derived by quinolin-2-one derivatives. The synthesized compounds were tested and screened in vitro against the urease-producing *R. mucilaginosa* and *Proteus mirabilis* strains. The results revealed that most of the tested compounds showed moderate-to-good activity [19]. Meanwhile, here we aim to synthesize another new series of quinolone-based 4-*O*-substituted-thiosemicarbazones and their 4-thiazolidinone derivatives, and explore them as antimicrobial and/or urease inhibitors (Figure 1). 

The intention to include quinolonyl moiety in our strategy was owing to its diverse range of biological properties, which include acetylcholinesterase inhibitor [20], antiallergenic [21], antimalarial [22], calcium-signaling inhibition [23] and antifungal [24] activities. Furthermore, quinolone hybrids have also been reported as potential candidates for antibacterial [25] and anticancer functions [26,27]. On the other hand, thiazolidinone ring has been linked to a variety of biological activities, including antibacterial [28], antitumor [29], antituberculous [30], and anti-inflammatory activities [31], and as potent urease inhibitors [32,33]. Furthermore, thiazolidinones are unique inhibitors of the bacterial enzyme MurB, a precursor involved in the biosynthesis of peptidoglycan as an essential component of both Gram-positive and Gram-negative bacterial cell wall [34,35,36].

## 2. Results and Discussion

### 2.1. Chemistry

The reaction sequences for synthesizing 4-thiazolidinones-quinolone hybrids **7a**–**l** starting from 4-hydroxyquinoline are outlined in Figure 1. The synthesis of ethyl 2-((2-oxo-1,2-dihydroquinolin-4-yl)oxy)acetate derivatives **3a**–**c** were obtained by refluxing ethyl bromoacetate (**2**) with 4-hydroxyquinoline **1a**–**c** in dry acetone in the presence of anhydrous potassium carbonate. For the synthesis of new 4-oxothiazolidin-quinolone hybrids **7a**–**l**, we planned to prepare the *N*,*N*-disubstituted thiourea derivatives **5a**–**g** as precursors for functionalized 4-oxothiazolidine derivatives. To approach these targets, the reaction of compounds **4a**–**c** and isothiocyanate derivatives in refluxing ethanol yielded the reported *N*,*N*-disubstituted thiourea derivatives **5a**,**b**,**d**–**f** [37]. All the newly synthesized compounds gave satisfactory analyses for the proposed structures, which were confirmed based on their IR, NMR, mass spectra, and elemental analyses. 

On the other hand, the structure of the newly prepared derivatives **5c** and **5g** were examined by elemental analyses, IR, and NMR in addition to mass spectra. For example, the ^1^H NMR spectrum of **5g** showed a singlet at *δ_N_* = 177.9 ppm assigned for N-4f. N-4f gives HMBC correlation with a singlet at *δ_H_* 5.29 and 5.36 ppm, assigned as H-4c and benzylic H-4i, respectively. Additionally, distinctive are the benzylic (2C-*m*), (2C-*o*), and (C-4i) at *δ_C_* = 128.48, 126.65, and 46.25 ppm, respectively. The distinctive carbons of **5g** are shown in Figure 2; for full correlations, please see Appendix A.

Interestingly, heterocyclization of 4-oxothiazolidine derivatives **7a**–**l** was carried out when hydrazinecarbothioamides **5a**–**g** were treated with dimethyl but-2-ynedioate (**6a**) and diethyl but-2-ynedioate (**6b**) in refluxing absolute ethanol for 6–8 h. The spectral and elemental data showed that series **7a**–**l** underwent the reaction smoothly to give the respective 4-oxothiazolidin-quinolone hybrid structure. The ^1^H NMR spectra revealed the disappearance of two NH signals. Additionally, the appearance of a new signal at ~160–163 ppm in ^13^C NMR spectra for new carbonyl thiazolidine moiety augments the formation of the thiazolidinone hybrid. As a representative example, the ^1^H NMR spectrum of compound **7b** (Table 1) showed quartet and triplet signals at *δ*_H_ = 4.27, 1.28 ppm for ethyl protons, as well as a singlet signal at *δ*_H_ = 6.91 ppm for (H-5a′); the benzylic protons (H-3a′) appeared as a singlet signal at *δ*_H_ = 4.69 ppm. Moreover, the ^13^C NMR spectrum (Table 1) showed signals at *δ_C_* = 160.94 ppm representing additional carbonyl group C-4′, and another signal of thiazolidin-4-one at *δ_C_* = 137.42 ppm is assigned as C-5′, which showed HMBC with *δ*_H_ = 6.91 and 4.69 ppm, assigned as H-5a′ and H-3a′, respectively. Further structural confirmation was also provided by analysis of the ^1^H-^15^N HSQC spectrum of **7b** (Table 1 and Appendix A), which showed a broad singlet at *δ*_H_ = 11.29 ppm, assigned as NH-4d, correlated with attached nitrogen, which appears at *δ*_N_ = 127.20 ppm. 

### 2.2. Bioactivities

#### 2.2.1. Antibacterial Screening

The *Proteus mirabilis* (*P. mirabilis*) strain was isolated from the urine of patients suffering from urinary tract infections. The strain was negative for hemolysis and motile and was urease positive. The test was performed using the cup-plate diffusion method [38] (Table 2). The results indicated that most tested compounds are weak or moderately active against *P. mirabilis*.

#### 2.2.2. Urease Inhibition Activity

Two series of quinolone derivatives containing thiosemicarbazone and thiazole functionalities **5a**–**g** and **7a**–**l** were synthesized and then screened for their in vitro urease inhibition properties using thiourea and acetohydroxamic acid as standard inhibitors [39]. The inhibition values of the synthesized compounds **5a**–**g** and **7a**–**l** exhibited moderate to good inhibitory effects when compared to clinically used enzyme inhibitors, thiourea and acetohydroxamic acid (Table 2 and Figure 3). It is worth mentioning that quinolone-thiosemicarbazone hybrids **5a**–**g** are effective molecules for urease inhibition ranking (IC_50_ = 1.83–11.21 µM) (Table 2 and Figure 3). Among the **5a**–**g** series, compounds **5a**–**c**, bearing a methyl group at position-1 on the quinolone skeleton, are the most active for urease inhibition (IC_50_ = 1.83–2.48 μM) when compared with the standard inhibitors thiourea and acetohydroxamic acid, which have IC_50_ values of 22.8 ± 1.31 and 21.03 ± 0.94 μM, respectively. Next, compounds **5d** and **5e** with *p*-methylquinolone moiety achieving IC_50_ = 5.29 ± 0.36 and 5.60 ± 0.84 μM were found to have superior activity compared to the standard inhibitors, thiourea and acetohydroxamic acid.

Meanwhile, **5f** and **5g**, with no substituents on quinolone moiety, showed the lowest activity (IC_50_ = 9.45 ± 0.08 and 11.21 ± 0.27 μM, respectively) compared to other members of the series, but were still more active than the standard inhibitors. It is clear that incorporating a methyl group at position-1 of the quinolone moiety enhances urease inhibitory activity. Overall, compound **5c** bearing a methyl group at position-1 and thiosemicarbazide phenyl terminal was proved to be the most potent inhibitor in the present series, achieving (IC_50_ = 1.83 ± 0.79 uM), as it exhibited relatively much greater activity, being approximately 12-fold more potent than thiourea and acetohydroxamic acid as references. On the other hand, the quinolone-thiazole hybrids **7a**–**l** were also screened in vitro for their human urease inhibitory potential. The results showed that most tested derivatives exhibited week to moderate urease inhibitory activities (IC_50_ = 18.80–45.43 µM). 

Among the **7a**–**l** series, compound **7c** exhibited good activity compared with the standard inhibitors, thiourea, and acetohydroxamic acid, achieving IC_50_ = 18.80 ± 1.72 μM (Table 2). From the above discussion, it can be concluded that the positions of substituents at the quinolone moiety and the thiosemicarbazone chain play vital roles in urease inhibitory activity. However, in silico docking was performed, and is discussed in the following paragraphs to verify these interpretations.

### 2.3. Molecular Docking

A molecular docking study was performed to elucidate the in vitro activities of all synthesized compounds and to understand their binding with the protein and subsequently figure out the most important pharmacophoric features in this scaffold. The docking protocol was performed by the Openeye scientific program against *bacillus pasteurii* urease (PDB: ID: 4ubp) [40,41,42]. 

The compounds’ interaction and their binding mode and pose are illustrated in Table 3. By analyzing the compound pose, the methylation of NH of quinolinyl moiety was very important because it hindered the quinolonyl part from forming HB with the amino acid clefts. Compound **5c** (*N*-phenylthiourea derivative) was the only compound whose thiourea functionality formed two HBs with the receptor in a chelation fashion (Figure 4a). To discover the great difference in activity between compounds **5a** and **5d**, overlay docking between these two ligands inside the receptor was performed. From Figure 4b, the quinolone ring in compound **5f** formed extra HB with Asp: 224A and one HB with Ala: 366A. Comparing compound **5c** with compound **5b** (benzylthiourea), it is clear that both compounds adopted different poses and modes with the receptor. However, compounds **5a** and **5b** illustrated similar poses (Figure 4c). These results indicate the electronic and geometric effect of the *N*-substituted thiourea part. 

The shape similarity for compounds **5a** and **5c** in comparison to lead 1 showed the quinolone ring overlaid with 4-F benzyl moiety (Figure 5). 

### 2.4. Structure–Activity Relationship (SAR)

Based on the compound activity and binding with the selected receptor, the series substituted thiourea has better activity than series b (thiazolidindione core). HB formation of the quinolonyl system is undesirable. Substitution on the phenyl of quinolone ring is ineffective. *N*-Phenylthiourea is the best derivative, and terminal substitution by alley or benzyl is very similar in terms of both activity and binding mode. These results open the gate for designing new derivatives bearing a substituted phenyl moiety. The latter is shown as in the case of compound **5c**, since the incorporation of a methyl group at position-1 of the quinolone moiety and thiosemicarbazide phenyl terminal enhances the urease inhibitory activity. Thus, compound **5c** exhibited relatively much greater activity, being approximately 12-fold more potent than thiourea and acetohydroxamic acid, as references. The same trend occurred in the case of thiazolidinone derivatives **7a**–**l**, as the electronic effect of the methyl groups in the quinolinone molecule, together with the presence of phenyl group (*N*-phenylthiazolidine) in compound **7c** (1,6-disubstituted-quinolinone-*N*-phenylthiazolidine), led the molecule to exhibit good activity compared with the standard inhibitors, thiourea, and acetohydroxamic acid.

## 3. Experimental Section

### 3.1. General Information

All reagents were purchased from Merck (St. Louis, MO, USA). The progress of all reactions was monitored with thin-layer chromatography (TLC) on Merck alumina-backed TLC plates and visualized under UV light. Spectra were measured in DMSO-*d_6_* on a Bruker AV-400 spectrometer (400 MHz for ^1^H, 100 MHz for ^13^C, and 40.54 MHz for ^15^N, in the Chemistry Department, Florida Institute of Technology, 150 W University Blvd, Melbourne, FL 32901, USA. Chemical shifts are expressed in δ (ppm) versus internal tetramethylsilane (TMS) = 0 ppm for ^1^H and ^13^C, and external liquid ammonia = 0 ppm for ^15^N. Coupling constants are stated in Hz. Correlations were established using ^1^H-^1^H COSY, and ^1^H-^13^C and ^1^H-^15^N HSQC and HMBC experiments. All ^15^N signals were observed indirectly via HSQC or HMBC experiments. Chemical shifts (δ) are reported in parts per million (ppm) relative to tetramethylsilane (TMS) as the internal standard, and the coupling constants (*J*) are reported in Hertz (Hz). Splitting patterns are denoted as follows: singlet (s), doublet (d), multiplet (m), triplet (t), quartet (q), doublet of doublets (dd), doublet of triplets (dt), triplet of doublets (td), and doublet of quartet (dq). Melting points (mp’s) were determined with a Stuart melting point instrument in the Chemistry Department, Florida Institute of Technology, 150 W University Blvd, Melbourne, FL, USA, and are expressed in °C. Mass spectra were recorded on a Finnigan Fab 70 eV at Al-Azhar University, Egypt. Elemental analyses were carried out on a Perkin Elmer device at the Microanalytical Institute of Organic Chemistry, Karlsruhe Institute of Technology, Karlsruhe, Germany.

### 3.2. Starting Materials

4-Hydroxyquinoline derivatives **1a**–**c** were prepared according to the literature [19,20]. Ethyl bromoacetate (**2**), isothiocyanate derivatives, dimethyl but-2-ynedioate (**6a**) and diethyl but-2-ynedioate (**6b**) were bought from Aldrich and used as received.


**General method for the synthesis of compounds 3a–c [37]**



**General method for the synthesis of compounds 4a–c [37]**



**General method for the synthesis of compounds 5a–e**


To a suspension of hydrazide derivatives **4a**–**c** (1 mmol) in absolute ethanol (30 mL), the appropriate isothiocyanate derivatives (1 mmol) were added, and the mixture was heated at reflux on a boiling water bath for 4–6 h. The mixture was then left to cool, and the precipitate so formed was collected by filtration, washed with methanol, and recrystallized from ethanol to give the target compounds **5a**–**e**.


***N*-Allyl-2-(2-((1-methyl-2-oxo-1,2-dihydroquinolin-4-yl)oxy)acetyl)hydrazinecarbo-thioamide (5a) [37]**


Yield: 0.270 g (78%), m.p. 177–179 °C. IR (KBr) υ_max_/cm^−1^ 3255, 3100.


***N*-Benzyl-2-(2-((1-methyl-2-oxo-1,2-dihydroquinolin-4-yl)oxy)acetyl)hydrazinecarbothioamide (5b) [37]**


Yield: 0.281 g (71%), m.p. 190–192 °C. IR (KBr) υ_max_/cm^−1^ 3249, 3150.


**2-(2-((1-Methyl-2-oxo-1,2-dihydroquinolin-4-yl)oxy)acetyl)-*N*-phenylhydrazinecarbothioamide (5c)**


Yield: 0.275 g (72%), m.p. 161–163 °C. IR (KBr) υ_max_/cm^−1^ 3214, 3110. **^1^H NMR:** 10.40 (bs, 1H; H-4e), 9.50 (bs, 1H; H-4f), 8.65 (b, 1H; H-4h), 8.11 (d, *J* = 7.9, 1H; H-5), 7.68 (dd, *J* = 7.7, 7.7, 1H; H-7), 7.53 (d, *J* = 8.5, 1H; H-8), 7.45 (dd, *J* = 8.0, 7.6, 2H; H-*m*), 7.30 (dd, *J* = 7.4, 7.4, 1H; H-6)), 7.25 (t, *J* = 7.4, 1H; H-*p*), 7.00 (dd, *J* = 8.5, 1.2, 2H; H-*o*), 6.04 (s, 1H; H-3), 4.86 (s, 2H; H-4c), 3.57 (s, 3H; H-1a). **^13^C NMR:** 165.41 (C-4d), 162.05 (C-2), 160.29 (C-4), 147.67 (C-*i*), 139.41 (C-8), 138.60 (C-8a), 131.55 (C-7), 129.57 (2C-*m*), 125.41 (C-*p*), 123.30 (C-5), 121.45 (C-6), 120.74 (2C-*o*), 115.24 (C-8), 114.56 (C-4a), 97.54 (C-3), 66.20 (C-4c), 28.66 (C-1a). **^15^N NMR:** 140.4 (N-1), 133.2 (N-4e/4f), 128.0 (N-4f/4e), 115.7 (N-4h). MS m/z (%): 382 (M^+^, 8), 257 (17), 132 (15), 65 (100). Anal. Calcd. for C_19_H_18_N_4_O_3_S (382.44): C, 59.67; H, 4.74; N, 14.65. Found: C, 59.82; H, 4.85; N, 14.83.


***N*-Allyl-2-(2-((6-methyl-2-oxo-1,2-dihydroquinolin-4-yl)oxy)acetyl)hydrazine-1-carbothioamide (5d) [37]**


Yield: 0.290 g (80%), m.p. 174–176 °C. IR (KBr) υ_max_/cm^−1^ 3289, 3165.


***N*-Benzyl-2-(2-((6-methyl-2-oxo-1,2-dihydroquinolin-4-yl)oxy)acetyl)hydrazinecarbothioamide (5e) [37]**


Yield: 0.297 g (75%), m.p. 183–185 °C. IR (KBr) υ_max_/cm^−1^ 3266, 3199. 


***N*-Allyl-2-(2-((2-oxo-1,2-dihydroquinolin-4-yl)oxy)acetyl)hydrazinecarbothioamide (5f) [37]**


Yield: 0.245 g (74%), m.p. 169–171 °C. IR (KBr) υ_max_/cm^−1^ 3234, 3120. 


***N*-Benzyl-2-(2-((2-oxo-1,2-dihydroquinolin-4-yl)oxy)acetyl)hydrazinecarbothioamide (5g)**


Yield: 0.271 g (71%), m.p. 199–201 °C. IR (KBr) υ_max_/cm^−1^ 3225, 3129. **^1^H NMR:** 14.16 (s, 1H; H-4e/4f/4h), 11.39 (bs, 1H; H-1), 7.44 (dd, *J* = 7.6, 7.6, 1H; H-7), 7.22 (m, 4H; H-*m, p,* 8), 7.18 (d, *J* = 7.7, 2H; H- *o*), 6.96 (d, *J* = 7.9, 1H; H-5), 6.91 (dd, *J* = 7.5, 7.5, 1H; H-6), 6.03 (s, 1H; H-3), 5.36 (s, 2H; H-4i), 5.29 (s, 2H; H-4c). **^13^C NMR:** 168.56 (C-4d), 162.83 (C-2), 161.00 (C-4), 147.36 (C-4g), 138.38 (C-8a), 135.50 (C-*i*), 130.90 (C-7), 128.48 (2C-*m*), 127.46 (C-*p*), 126.65 (2C-*o*), 122.21 (C-5), 121.16 (C-6), 114.87 (C-8), 113.84 (C-4a), 97.97 (C-3), 60.26 (C-4c), 46.25 (C-4i). **^15^N NMR:** 282.4 (N-4e), 177.9 (N-4f), 144.2 (N-1). MS m/z (%): 382 (M^+^, 28), 280 (100), 188 (34), 47 (33). Anal. Calcd. for C_19_H_18_N_4_O_3_S (382.44): C, 59.67; H, 4.74; N, 14.65. Found: C, 59.85; H, 4.97; N, 14.83


**General method for the synthesis of compounds 7a–l**


To a solution of thiosemicarbazide **5a**–**g** (1 mmol) in abs. ethanol (25 mL), **DMAD** (**6a**), and **DEAD** (**6b**) (0.17 gm, 1 mmol) were added, and the mixture was heated under reflux for 8–10 h. The mixture was then left to cool. The formed precipitate was filtered off, washed with hot ethanol, and recrystallized from methanol to give the target compounds **7a**–**l**.


**(2*E*)-Methyl 2-(3-allyl-2-(2-(2-((1-methyl-2-oxo-1,2-dihydroquinolin-4-yl)oxy)acetyl)-hydrazono)-4-oxothiazolidin-5-ylidene)acetate (7a)**


Yield: 0.342 g (75%), m.p. 233–235 °C. IR (KBr) υ_max_/cm^−1^ 1735, 1630, 1051. **^1^H NMR:** 7.81 (d, *J* = 7.8, 1H; H-5), 7.66 (dd, *J* = 8.2, 7.4, 1H; H-7), 7.53 (d, *J* = 8.5, 1H; H-8), 7.26 (dd, *J* = 7.5, 7.5, 1H; H-6), 6.77 (s, 1H; H-5a′), 6.31 (s, 1H; H-3), 5.99 (ddt, *J*_d_ = 17.0, 10.2, *J*_t_ = 5.2, 1H; H-3b′), 5.52 (s, 2H; H-4b), 5.18 (d, *J* = 10.3, 1H; H-3c′), 5.00 (d, *J* = 17.4, 1H; H-3c′), 4.86 (d, *J* = 4.2, 2H; H-3a′), 3.78 (s, 3H; H-5c′), 3.57 (s, 3H; H-1a), 3.48 (s; NH-1). **^13^C NMR:** 164.84 (C-5b′), 163.11 (C-4′), 162.00 (C-2), 159.84 (C-4), 151.49 (C-4c), 148.25 (C-5a′), 139.42 (C-8a), 132.11 (C-3b′), 131.57 (C-7), 122.78 (C-5a′), 122.56 (C-5), 121.60 (C-6), 117.80 (C-3c′), 115.10 (C-8), 114.74 (C-4a), 97.95 (C-3), 60.52 (C-4b), 52.37 (C-5c′), 46.69 (C-3a′), 28.67 (C-1a). **^15^N NMR:** 329.1 (N-4e), 177.1 (N-4d), 137.6 (N-1). N-3′ n/o. MS m/z (%): 456 (M^+^, 38), 392 (62), 148 (38), 44 (100). Anal. Calcd. for C_21_H_20_N_4_O_6_S (456.47): C, 55.26; H, 4.42; N, 12.27. Found C, 55.47; H, 4.59; N, 12.58.


**(2*E*)-Methyl 2-(3-allyl-2-(2-(2-((1-methyl-2-oxo-1,2-dihydroquinolin-4-yl)oxy)acetyl)-hydrazono)-4-oxothiazolidin-5-ylidene)acetate (7b)**


Yield: 0.405 g (77%), m.p. 243–245 °C. IR (KBr) υ_max_/cm^−1^ 1738, 1640, 1054. **^1^H NMR:** 11.29 (s, 1H; NH-4d), 8.10 (dd, *J* = 8.0, 1.0, 1H; H-5), 7.68 (ddd, *J* = 7.2, 7.2, 1.1, 1H; H-7), 7.53 (d, *J* = 8.5, 1H; H-8), 7.30 (m, 6H; H-*o, m, p,* 6), 6.91 (s, 1H; H-5a′), 6.12 (s, 1H; H-3), 5.03 (s, 2H; H-4b), 4.69 (s, 2H; H-3a′), 4.27 (q, *J* = 7.1, 2H; H-5c′), 3.57 (s, 3H; H-1a), 1.28 (t, *J* = 7.1, 3H; H-5d′). **^13^C NMR:** 165.61 (C-4c), 165.01 (C-5b′), 162.00 (C-2), 160.94 (C-4′), 160.00 (C-4), 145.89 (C-2′), 139.42 (C-8a), 138.26 (C-*i*), 137.42 (C-5′), 131.57 (C-7), 128.30 (2C-*m*), 127.40 **(**2C-*o*), 126.91 (C-*p*)*,* 123.24 (C-5), 121.43 (C-6), 116.97 (C-5a′), 115.15 (C-4a), 114.58 (C-8), 97.92 (C-3), 66.06 (C-4b), 61.63 (C-5c′), 54.76 (C-3a′), 28.67 (C-1a), 13.94 (C-5d′). **^15^N NMR:** 137.9 (N-1), 127.2 (N-4d). N-3′, 4e n/o. MS m/z (%): 520 (M^+^, 77), 236 (93), 200 (100), 40 (71). Anal. Calcd. for C_26_H_24_N_4_O_6_S (520.56): C, 59.99; H, 4.65; N, 10.76. Found C, 60.12; H, 4.82; N, 10.89.


**(2*E*)-Methyl 2-(2-(2-(2-((1-methyl-2-oxo-1,2-dihydroquinolin-4-yl)oxy)acetyl)-hydrazono)-4-oxo-3-phenylthiazolidin-5-ylidene)acetate (7c)**


Yield: 0.400 g (81%), m.p. 229–231 °C. IR (KBr) υ_max_/cm^−1^ 1740, 1634, 1060. **^1^H NMR:** 11.47 (b, 1H; NH-4d), 8.15 (d, *J* = 8.0, 1H; H-5), 7.69 (ddd, *J* = 8.5, 7.1, 1.4, 1H; H-7), 7.54 (d, *J* = 8.4, 1H; H-8), 7.45 (dd, *J* = 8.0, 7.6, 2H; H-*m*), 7.31 (dd, *J* = 7.8, 7.3, 1H; H-6), 7.25 (t, *J* = 7.4, 1H; H-*p*), 7.00 (dd, *J* = 8.3, 0.9, 2H; H-*o*), 6.97 (s, 1H; H-5a′), 6.11 (s, 1H; H-3), 5.09 (s, 2H; H-4b), 3.77 (s, 3H; H-5c′), 3.57 (s, 3H; H-1a). **^13^C NMR:** 165.60 (C-4c), 165.06 (C-5b′), 162.00 (C-2), 160.92 (C-4′), 160.03 (C-4), 147.57 (C-*i*), 146.44 (C-2′), 139.43 (C-8a), 137.56 (C-5′), 131.60 (C-7), 129.58 **(**2C-*m*), 125.43 (C-*p*), 123.28 (C-5), 121.48 (C-6), 120.73 **(**2C-*o*), 117.65 (C-5a′), 115.15 (C-8), 114.61 (C-4a), 97.93 (C-3), 66.06 (C-4b), 52.71 (C-5c′), 28.68 (C-1a). **^15^N NMR:** 137.7 (N-1). MS m/z (%): 492 (M^+^, 29), 204 (100), 145 (49), 40 (22). Anal. Calcd. for C_24_H_20_N_4_O_6_S (492.50): C, 58.53; H, 4.09; N, 11.38. Found C, 58.67; H, 4.27; N, 11.56.


**(2*E*)-Ethyl 2-(2-(2-(2-((1-methyl-2-oxo-1,2-dihydroquinolin-4-yl)oxy)acetyl)-hydrazono)-4-oxo-3-phenylthiazolidin-5-ylidene)acetate (7d)**


Yield: 0.399 g (78%), m.p. 223–225 °C. IR (KBr) υ_max_/cm^−1^ 1736, 1639, 1077. **^1^H NMR:** 11.47 (bs, 1H; NH-4d), 8.15 (dd, *J* = 8.0, 1.3, 1H; H-5), 7.68 (ddd, *J* = 8.5, 7.2, 1.4, 1H; H-7), 7.53 (d, *J* = 8.5, 1H; H-8), 7.45 (dd, *J* = 8.0, 7.6, 2H; H-*m*), 7.31 (ddd, *J* = 7.9, 7.3, 0.6, 1H; H-6), 7.25 (t, *J* = 7.4, 1H; H-*p*), 7.00 (dd, *J* = 8.5, 1.2, 2H; H-*o*), 6.94 (s, 1H; H-5a′), 6.11 (s, 1H; H-3), 5.09 (s, 2H; H-4b), 4.22 (q, *J* = 7.1, 2H; H-5c′), 3.57 (s, 3H; H-1a), 1.24 (t, *J* = 7.1, 3H; H-5d′). **^13^C NMR**: 165.60 (C-4c), 165.06 (C-5b′), 162.00 (C-2), 160.92 (C-4′), 160.03 (C-4), 147.67 (C-*i*), 146.45 (C-2′), 139.42 (C-8a), 137.53 (C-5′), 131.60 (C-7), 129.57 (2C-*m*), 125.41 (C-*p*), 123.28 (C-5), 121.48 (C-6), 120.74 (2C-*o*), 117.65 (C-5a′), 115.15 (C-8), 114.60 (C-4a), 97.92 (C-3), 66.06 (C-4b), 61.71 (C-5c′), 28.68 (C-1a), 13.87 (C-5d′). **^15^N NMR:** 265.2 (N-3′), 137.7 (N-1), 127.4 (N-4d). N-4e n/o. MS m/z (%): 506 (M^+^, 54), 316 (58), 181 (100), 45 (26). Anal. Calcd. for C_25_H_22_N_4_O_6_S (506.53): C, 59.28; H, 4.38; N, 11.06. Found C, 59.39; H, 4.62; N, 11.31.

**(*E*)-Ethyl 2-((*Z*)-3-allyl-2-(2-(2-((6-methyl-2-oxo-1,2-dihydroquinolin-4-yl)oxy)acetyl)-hydrazono)-4-oxothiazolidin-5-ylidene)acetate** (**7e**)

Yield: 0.375 g (79%), m.p. 239–241 °C. IR (KBr) υ_max_/cm^−1^ 1744, 1640, 1074.**^1^H NMR:** 11.35 (b, 1H; NH-1), 11.13 (bs, 1H; NH-4d), 7.71 (s, 1H; H-5), 7.37 (d, *J* = 8.3, 1H; H-7), 7.20 (d, *J* = 8.4, 1H; H-8), 6.81 (s, 1H; H-5a′), 5.89 (ddt, *J*_d_ = 17.2, 9.9, *J*_t_ = 5.2, 1H; H-3b′), 5.83 (s, 1H; H-3), 5.19 (d, *J* = 10.4, 1H; H-3c′), 5.15 (d, *J* = 17.4, 1H; H-3c′), 4.88 (s, 2H; H-4b), 4.43 (m, 2H; H-3a′), 4.27 (q, *J* = 7.1, 2H; H-5c′), 2.37 (s, 3H; H-6a), 1.27 (t, *J* = 6.7, 3H; H-5d′). **^13^C NMR:** 165.32 (C-5b′), 163.27 (C-4c, 4′), 162.86 (C-2), 161.62 (C-4), 151.5 (C-2′), 140.17 (C-5′), 136.66 (C-8a), 132.24 (C-7), 130.73 (C-3b′), 130.37 (C-6), 122.07 (C-5), 117.64 (C-3c′), 115.58 (C-5a′), 115.10 (C-8), 114.24 (C-4a), 97.56 (C-3), 66.15 (C-4b), 61.57 (C-5c′), 44.60 (C-3a′), 20.48 (C-6a), 13.96 (C-5d′). **^15^N NMR:** 157.0 (N-4d), 143.8 (N-1). MS m/z (%): 470 (M^+^, 18), 338 (53), 106 (100), 40 (16). Anal. Calcd. for C_22_H_22_N_4_O_6_S (470.50): C, 56.16; H, 4.71; N, 11.91. Found C, 56.37; H, 4.89; N, 12.07.


**(*E*)-Methyl 2-((*Z*)-3-allyl-2-(2-(2-((6-methyl-2-oxo-1,2-dihydroquinolin-4-yl)oxy)-acetyl)hydrazono)-4-oxothiazolidin-5-ylidene)acetate (7f)**


Yield: 0.342 g (75%), m.p. 232–233 °C. IR (KBr) υ_max_/cm^−1^ 1780, 1666, 1090. **^1^H NMR**: 11.35 (s, 1H; NH-1), 11.15 (bs, 1H; NH-4d), 7.71 (s, 1H; H-5), 7.37 (d, *J* = 8.2, 1H; H-7), 7.20 (d, *J* = 8.3, 1H; H-8), 6.83 (s, 1H; H-5a′), 5.89 (ddt, *J*_d_ = 17.1, 10.2, *J*_t_ = 5.0, 1H; H-3b′), 5.83 (s, 1H; H-3), 5.19 (d, *J* = 11.2, 1H; H-3c′), 5.15 (d, *J* = 17.1, 1H; H-3c′), 4.88 (s, 2H; H-4b), 4.43 (m, 2H; H-3a′), 3.81 (s,3H; H-5c′), 2.37 (s, 3H; H-6a). **^13^C NMR**: 165.78 (C-5b′), 163.28 (C-4c), 163.08 (C-4′), 162.86 (C-2), 161.62 (C-4), 151.28 (C-2′), 140.22 (C-5′), 136.66 (C-8a), 132.25 (C-7), 130.73 (C-3b′), 130.37 (C-6), 122.05 (C-5), 117.66 ( C-3c′), 115.32 (C-8), 115.10 (C-5a′), 114.22 (C-4a), 97.56 (C-3), 66.13 (C-4b), 52.63 (C-5c′), 44.64 (C-3a′), 20.48 (C-6a). **^15^N NMR**: 157.0 (N-4d), 143.4 (N-1); N-4e, 3′ n/o. MS m/z (%): 456 (M^+^, 22), 360 (46), 216 (100), 43 (45). Anal. Calcd. for C_21_H_20_N_4_O_6_S (456.47): C, 55.26; H, 4.42; N, 12.27. Found C, 55.49; H, 4.60; N, 12.53.


**(2*E*)-Methyl 2-(3-benzyl-2-(2-(2-((6-methyl-2-oxo-1,2-dihydroquinolin-4-yl)oxy)acetyl)-hydrazono)-4-oxothiazolidin-5-ylidene)acetate (7g)**


Yield: 0.355 g (70%), m.p. 241–243 °C. IR (KBr) υ_max_/cm^−1^ 1799, 1694, 1041. **^1^H NMR:** 11.36 (s, 1H; NH-1), 11.25 (s, 1H; NH-4d), 7.77 (s, 1H; H-5), 7.31 (m, 6H; H-7, *o, m, p*), 7.23 (d, *J* = 8.3, 1H; H-8), 6.86 (s, 1H; H-5a′), 5.81 (s, 1H; H-3), 5.90 (s, 2H; H-3a′), 4.99 (s, 2H; H-4b), 3.81 (s, 3H; H-5c′), 2.34 (s, 3H; H-6a). **^13^C NMR:** 165.6 (C-5b′), 165.4 (C-4c), 162.7 (C-4′) 161.1 (C-2), 160.9 (C-4), 145.8 (C-2′), 138.2 (C-5′), 137.5 (C-*i*), 136.6 (C-8a), 132.2 (C-7), 130.2 (C-6), 128.3 (5C-*o, m, p*), 126.9 (C-5), 122.0 (C-5a′), 116.7 (C-8), 114.1 (C-4a), 98.2 (C-3), 66.0 (C-4b), 54.7 (C-5c′), 52.6 (C-3a′), 20.47 (C-6a). MS m/z (%): 506 (M^+^, 22), 377 (33), 283 (100), 57 (20). Anal. Calcd. for C_25_H_22_N_4_O_6_S (506.53): C, 59.28; H, 4.38; N, 11.06. Found C, 59.64; H, 4.51; N, 11.34.

**(*E*)-Ethyl 2-((*Z*)-3-benzyl-2-(2-(2-((6-methyl-2-oxo-1,2-dihydroquinolin-4-yl)oxy)acetyl)-hydrazono)-4-oxothiazolidin-5-ylidene)acetate** (**7h**)

Yield: 0.390 g (75%), m.p. 233–235 °C. IR (KBr) υ_max_/cm^−1^ 1756, 1660, 1083. **^1^H NMR:** 11.35 (s, 1H; NH-1), 11.17 (s, 1H; NH-4d), 7.71 (s, 1H; H-5), 7.3 (m, 6H; H-7, *o, m, p*), 7.20 (d, *J* = 8.3, 1H; H-8), 6.83 (s, 1H; H-5a′), 5.83 (s, 1H; H-3), 5.01 (s, 2H; H-3a′), 4.88 (s, 2H; H-4b), 4.27 (q, *J* = 6.9, 2H; H-5c′), 2.37 (s, 3H; H-6a), 1.27 (t, *J* = 7.1, 3H; H-5d′). **^13^C NMR:** 165.6 (C-5b′), 163.7 (C-4c), 163.3 (C-4′), 162.5 (C-2), 162.4 (C-4), 151.6 (C-2′), 137.7 (C-*i*), 137.41 (C-5′), 136.8 (C-8a), 132.7 (C-7), 130.7 (C-6), 128.6 (5C-*o, m, p*), 122.4 (C-5), 116.0 (C-5a′), 115.4 (C-8), 114.9 (C-4a), 98.0 (C-3), 66.4 (C-4b), 62.0 (C-5c′), 46.1 (C-3a′), 20.9 (C-6a), 14.4 (C-5d′). MS m/z (%): 520 (M^+^, 17), 428 (14), 91 (100), 40 (39). Anal. Calcd. for C_26_H_24_N_4_O_6_S (520.56): C, 59.99; H, 4.65; N, 10.76. Found C, 60.13; H, 4.81; N, 10.98.


**(2*E*)-Ethyl 2-(3-allyl-4-oxo-2-(2-(2-((2-oxo-1,2-dihydroquinolin-4-yl)oxy)acetyl)hydra-zono)thiazolidin-5-ylidene)acetate (7i)**


Yield: 0.360 g (78%), m..p. 210–212 °C. IR (KBr) υ_max_/cm^−1^ 1741, 1639, 1041. **^1^H NMR:** 11.44 (b, 1H), 11.15 (b, 1H; NH-1, 4d), 7.92 (d, *J* = 7.9, 1H; H-5), 7.54 (dd, *J* = 7.8, 7.5, 1H; H-7), 7.30 (d, *J* = 8.2, 1H; H-8), 7.20 (dd, *J* = 7.6, 7.5, 1H; H-6), 6.80 (s, 1H; H-5a′), 5.88 (ddt, *J*_d_ = 17.3, 10.3, *J*_t_ = 5.2, 1H; H-3b′), 5.85 (s, 1H; H-c), 5.28 (d, *J* = 17.2, 1H; H-3c′), 5.20 (m, 1H; H-3c′), 4.90 (s, 2H; H-4b), 4.43 (m, 2H; H-3a′), 4.27 (q, *J* = 7.1, 2H; H-5c′), 1.27 (t, *J* = 6.9, 3H; H-5d′). **^13^C NMR:** 165.32 (C-5b′), 163.25 (C-4′), 163.08 (C-2′), 162.97 (C-2), 161.77 (C-4), 151.54 (C-4c), 140.12 (C-5′), 138.64 (C-8a), 131.09 (C-7), 130.72 (C-3b′), 122.63 (C-5), 121.34 (C-6), 117.63 (C-3c′), 115.57 (C-5a′), 115.14 (C-8), 114.34 (C-4a), 97.61 (C-3), 66.19 (C-4b), 61.57 (C-5c′), 44.61 (C-3a′), 13.97 (C-5d′). **^15^N NMR:** 156.8, (N-1) 144.2 (N4d). N-3′, 4e n/o. MS m/z (%): 456 (M^+^, 19), 320 (30), 129 (100), 40 (13). Anal. Calcd. for C_21_H_20_N_4_O_6_S (456.47): C, 55.26; H, 4.42; N, 12.27. Found C, 55.43; H, 4.57; N, 12.45.


**(2*E*)-Methyl 2-(3-allyl-4-oxo-2-(2-(2-((2-oxo-1,2-dihydroquinolin-4-yl)oxy)acetyl)-hydrazono)thiazolidin-5-ylidene)acetate (7j)**


Yield: 0.330 g (73%), m.p. 244–246 °C. IR (KBr) υ_max_/cm^−1^ 1794, 1680, 1084. **^1^H NMR:** 11.43 (bs, 1H; N**H**-1), 11.21 (b, 1H; N**H**-4d), 7.92 (d, *J* = 7.9, 1H; H-5), 7.54 (dd, *J* = 7.7, 7.6, 1H; H-7), 7.30 (d, *J* = 8.2, 1H; H-8), 7.20 (dd, *J* = 7.6, 7.4, 1H; H-6), 6.83 (s, 1H; H-5a′), 5.90 (ddt, *J*_d_ = 16.7, 10.9, *J*_t_ = 5.4, 1H; H-3b′), 5.85 (s, 1H; H-3), 5.19 (d, *J* = 11.4, 1H; H-3c′), 5.17 (d, *J* = 16.7, 1H; H-3c′), 4.90 (s, 2H; H-4b), 4.44 (m, 2H; H-3a′), 3.81 (s, 3H; H-5c′). **^13^C NMR:** 165.78 (C-5b′), 163.28 (C-2′, 4′), 162.98 (C-2), 161.78 (C-4), 151.34 (C-4c), 140.27 (C-5′), 138.64 (C-8a), 131.27 (C-7), 131.09 (C-3b′), 122.62 (C-5), 121.34 (C-6), 117.65 (C-3c′), 115.30 (C-8), 115.14 (C-5a′), 114.35 (C-4a), 97.61 (C-3), 66.23 (C-4b), 52.63 (C-5c′), 44.65 (C-3a′). **^15^N NMR:** 144.2 (N-1); N-3′, N-4d, N-4e n/o. MS m/z (%): 442 (M^+^, 33), 360 (63), 283 (100), 41 (13). Anal. Calcd. for C_20_H_18_N_4_O_6_S (442.45): C, 54.29; H, 4.10; N, 12.66. Found C, 54.51; H, 4.28; N, 12.92


**(2*E*)-Methyl 2-(3-benzyl-4-oxo-2-(2-(2-((2-oxo-1,2-dihydroquinolin-4-yl)oxy)acetyl)-hydrazono)thiazolidin-5-ylidene)acetate (7k)**


Yield: 0.375 g (76%), m.p. 251–253 °C. IR (KBr) υ_max_/cm^−1^ 1781, 1683, 1092. **^1^H NMR:** 11.42 (bs, 1H; N**H**-1), 11.19 (b, 1H; N**H**-4d), 7.91 (d, *J* = 7.9, 1H; H-5), 7.44 (dd, *J* = 7.6, 7.5, 1H; H-7), 7.32 (m, 4H; H-*o, m*), 7.32 (m, 1H; H-*p*), 7.29 (d, *J* = 8.5, 1H; H-8), 7.20 (dd, *J* = 7.4, 6.8, 1H; H-6), 6.80 (s, 1H; H-5a′), 5.83 (s, 1H; H-3), 5.11 (s, 2H; H-4b), 4.95 (s, 2H; H-3a′), 4.30 (s, 3H; H-5c′). **^13^C NMR:** 165.77 (C-5b′), 163.38 (C-4c), 162.98 (C-2), 161.45 (2C-4, 4′), 151.48 (C-2′), 138.64 (2C-8a, *i*), 135.35 (C-5′), 131.08 (C-7), 128.50 (C-*p*), 128.33 (2C-*o*),, 127.98 (2C-*m*), 127.73 (C-5), 127.42 (C-6), 121.34 (C-5a′), 115.14 (2C-8, 4a), 97.64 (C-3), 66.16 (C-3a′), 61.59 (C-5c′), 52.61 (C-4b). MS m/z (%): 492 (M^+^, 25), 375 (20), 343 (100), 147 (38). Anal. Calcd. for C_24_H_20_N_4_O_6_S (492.51): C, 58.53; H, 4.09; N, 11.38. Found C, 58.64; H, 4.28; N, 11.56.


**(*E*)-Ethyl 2-((*Z*)-3-benzyl-4-oxo-2-(2-(2-((2-oxo-1,2-dihydroquinolin-4-yl)oxy)acetyl)-hydrazono)thiazolidin-5-ylidene)acetate (7l)**


Yield: 0.370 g (72%), m.p. 249–251 °C. IR (KBr) υ_max_/cm^−1^ 1790, 1689, 1088. **^1^H NMR**: 11.44 (bs, 1H; NH-1), 11.22 (b, 1H; NH-4d), 7.92 (d, *J* = 7.9, 1H; H-5), 7.54 (dd, *J* = 7.6, 7.5, 1H; H-7), 7.39 (m, 4H; H-*o*, *m*), 7.34 (m, 1H; H-*p*), 7.30 (d, *J* = 8.5, 1H; H-8), 7.20 (dd, J = 7.4, 6.8, 1H; H-6), 6.82 (s, 1H; H-5a′), 5.86 (s, 1H; H-3), 5.01 (s, 2H; H-4b), 4.90 (s, 2H; H-3a′), 4.27 (q, *J* = 7.0, 2H; H-5c′), 1.27 (t, *J* = 7.0, 3H; H-5d′). **^13^C NMR**: 165.31 (C-5b′), 163.38 (C-4c), 162.99 (C-2), 161.75 (2C-4, 4′), 151.48 (C-2′), 138.64 (2C-8a, *i*), 135.35 (C-5′), 131.09 (C-7), 128.49 (C-p), 127.96 (2C-*o*), 127.73 (2C-*m*), 122.64 (C-5), 121.34 (C-6), 115.82 (C-5a′), 115.14 (2C-8, 4a), 97.62 (C-3), 66.16 (C-3a′), 61.59 (C-5c′), 45.83 (C-4b), 13.96 (C-5d′). **^15^N NMR**: 144.2 (N-1). MS m/z (%): 506 (M^+^, 57), 372 (37), 232 (45), 113 (100). Anal. Calcd. for C_25_H_22_N_4_O_6_S (506.53): C, 59.28; H, 4.38; N, 11.06. Found C, 59.45; H, 4.57; N, 11.32.

### 3.3. Biology

#### Urease Inhibitory Activity

*In vitro* screening and inhibitory studies on urease (Jack bean urease) were determined using the colored Berthelot phenols method, which measures the liberation of ammonia from the reaction [43]. Briefly, the assay mixture containing 1 unit of the enzyme was added to 650 µL of buffer solution (50 mmol phosphate buffer Ph 6.7, 400 mmol sodium salicylate, 10 mmol sodium nitroprusside, and 2 mmol EDTA/L) and mixed with 10 µL of different concentration 0.1–100 Mm of the tested compounds in DMF as a solvent. DMF was tested alone and showed no inhibitory effect on the enzyme. After 15 min incubation at room temperature, 10 µL of 50 mg/L urea solution was added. The mixture was incubated for 0.5 h in a water bath at 37 °C to allow the hydrolysis process. 

After complete urea hydrolysis and ammonia liberation, the reaction was stopped by adding 200 µL of the hypochlorite reagent (150 mmol/L sodium hydroxide, 140 mm/L sodium hypochlorite). The liberated ammonia was allowed to complex with the hypochlorite and salicylate for 25 min at 30 °C. The absorbance was measured at 578 nm using a UV/VIS Spectrophotometer (Optizen POP, 5U4608, Daejeon, Korea), and experiments were performed in triplicate in a final volume of 1 mL. All results were compared with thiourea, a standard inhibitor of urease. The percentage inhibition was calculated as the difference of absorbance values with and without the test compounds. The concentration that provokes an inhibition halfway between the minimum and maximum response of each compound (relative IC_50_) was determined by monitoring the inhibition effect of various concentrations of compounds in the assay.

## 4. Molecular Docking Study

A docking study was performed for the target compounds using the Openeye scientific program (academic license 2021). The coordinate for the protein structure was obtained from the Protein Data Bank (PDB ID: 4ubp). The compound conformers were energy minimized using the Omega application. The docking step was operated by the Fred application, and the results were visualized by the Vida command.

## 5. Conclusions

Thiazolidinone derivatives were achieved starting from 4-hydroxyquinolin-2-ones, which were subjected to ethyl bromoacetate to afford the corresponding ethyl oxoquinolinyl acetates. The latter species reacted with hydrazine hydrate to afford the hydrazide derivatives. Then, the hydrazide derivatives reacted with isothiocyanate derivatives to give the corresponding *N*,*N*-disubstituted thioureas. On subjecting the *N*,*N*-disubstituted thioureas with dialkyl acetylenedicarboxylates, the thiazolidinone derivatives was obtained in good yields. The two series based on quinolone ring, one bearing *N*,*N*-substituted thiourea, and the other bear thiazolidinone ring were designed with different substituents at different positions. The *N*,*N*-disubstituted thiourea scaffold with *N*-methyl quinolone system exhibited the most potent urease inhibitor activity. Besides the study of the previous results dealing with the results of urease inhibition activity of 4-*O*-substituted-thiosemicarbazones and derived by quinolin-2-ones, it can be concluded that the quinolinone moiety plays an important role in the mechanism of urease inhibition process. 

## Data Availability

Not applicable.

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
