# Peer review of "Synthesis and Identification of New N,N-Disubstituted Thiourea, and Thiazolidinone Scaffolds Based on Quinolone Moiety as Urease Inhibitor"

_molecules, 2022, doi:10.3390/molecules27207126_

Round 1

Reviewer 1 Report

I would like to ask the authors to carefully check the English in the article, because moderate English correction is required. However, the main drawback of this work, in my opinion, is the lack of sufficient information about the novelty and advantages of the results obtained compared to the known data on this topic. It is important to point out more clearly what are the differences and advantages of the new results obtained in comparison with the known data, including the results obtained by the authors earlier. It is also necessary to formulate the conclusions more clearly in the sections "Abstract" and "5. Conclusion".

In general, the data obtained are useful information for chemists working in this field of research and deserve publication. I would recommend the authors to correct the article taking into account the above comments. 

Author Response

We recheck language mistakes and correct mistakes. Rephrase cloudy phrases to be more clear

However, the main drawback of this work, in my opinion, is the lack of sufficient information about the novelty and advantages of the results obtained compared to the known data on this topic.

In introduction:

We added the sentence" That previous strategy was involved the synthesis of 3-thiosemicarbazides derived by quinolin-2-one derivatives. (lines 77-85). Some sentences were corrected in order to show the results of the previous work and the comparsion between the structure of our previous ones all in red highlights.

It is important to point out more clearly what are the differences and advantages of the new results obtained in comparison with the known data, including the results obtained by the authors earlier.

It is also necessary to formulate the conclusions more clearly in the sections "Abstract" and "5. Conclusion".

Look at the abstract, we added a new "The inhibition values of the synthesized compounds thioureas and thiazolidinones exhibited moderate to good inhibitory effects".

In general lines 34-40 were added to the abstract to make the results more clear to the reader (red highlight)

In conclusion: a paragraph was added started by Comparsion study………in lines 491-493 was added (red highlight).

In general, the data obtained are useful information for chemists working in this field of research and deserve publication. I would recommend the authors to correct the article taking into account the above comments.

Was done in the introduction, abstract and conclusion.

Reviewer 2 Report

Overall, the study is well done and can be recommended for publication in the Molecules, but much work is needed to improve the manuscript (see below).

The purpose of the study is to obtain substances with two fragments with a potential urease inhibitory activity is quite clear. However, its perception is greatly hindered by "garbage" references (Ref. 9-19 and 23-40, 43-45) to other biological activity of both fragments, which has nothing to do with this study.

The sentence on lines 57-59 is not very clear - what is meant by "agricultural applications"?

It would be worth giving a name to the compound lead 1 in Figure 1.

The text in lines 91-97 is identical to the text of the reference 47. However, it is misleading, since it gives the impression that these are new substances, while these compounds have already been obtained previously. There is also no clear indication on page 4 that compounds 5a,b and 5d-f are not just "known" compounds, but they were obtained by the authors in a previous publication (Ref. 47) according to Scheme 1. In Scheme 1, designations R and R' should be given for compounds 4a-c. It is also worth noting that substances 5a-d are not at all "key intermediates", as indicated in line 104, but one of the main compounds stated in the title of the manuscript and, moreover, having the highest biological activity.

It is very strange to see a description of NMR spectra (including two-dimensional ones) without pictures of the spectra themselves. Pictures of spectra, especially two-dimensional ones (on the basis of which the signal assignments were made) should be given at least in the Supporting Information, which is not available at all. This is more illustrative and gives the reader much more information than, for example, Table 1 (which can be deleted), especially since the assignment of signals in the spectra is presented in the experimental part.

I would recommend using the more commonly used names for compounds 6a and 6b: dimethyl and diethyl acetylene dicarboxylate, respectively.

Line 170 – Reference 49 is absent.

The caption to Figure 3 is unacceptable - here you should indicate what IC50 is and give the numbers of the compounds in the figure itself.

Line 197 – Table 4?? Probably it should be Table 2.

Figure 4 and Table 2 duplicate each other and one of them (figure) can be deleted.

Section 2.4 (Structure-Activity Relationship) is very short and needs to be expanded. Did the authors consider the possibility of synthesizing substances with two methyl groups in positions 1 and 6 of the quinolone fragment?

In Experimental, Section 3.1 (Material and Methods) is empty. Sections 3.4 and 3.5 should be combined with Section 3.3 (Starting Compounds). The source of 6a and 6b should be indicated here as well.

Author Response

  • Reviewer 2

    • The purpose of the study is to obtain substances with two fragments with a potential urease inhibitory activity is quite clear. However, its perception is greatly hindered by "garbage" references (Ref. 9-19 and 23-40, 43-45) to other biological activity of both fragments, which has nothing to do with this study.

    Thanks for your comment but we give a general data in the introduction part about the importance and the biological activity of each moiety in the targeted compounds (quinolone, thiazolidinone and thiourea) so these moieties represent a broad spectrum biologically active moieties and the targeted compounds may be further studied for other biological activities in the near future.

    • The sentence on lines 57-59 is not very clear - what is meant by "agricultural applications"?

    The use of urea-based fertilizers has negative consequences in the form of ammonia volatilization. The solution to this problem may be the use of urease inhibitors.

    • It would be worth giving a name to the compound lead 1 in Figure 1.

    We add name of Lead 1

    • The text in lines 91-97 is identical to the text of the reference 47. However, it is misleading, since it gives the impression that these are new substances, while these compounds have already been obtained previously. There is also no clear indication on page 4 that compounds 5a,b and 5d-f are not just "known" compounds, but they were obtained by the authors in a previous publication (Ref. 47) according to Scheme 1. In Scheme 1, designations R and R' should be given for compounds 4a-c. It is also worth noting that substances 5a-d are not at all "key intermediates", as indicated in line 104, but one of the main compounds stated in the title of the manuscript and, moreover, having the highest biological activity.

    We re-phrase lines 113-118 to illustrate that compounds 5a,b,d-f were synthesized by our group and reported in ref 47 but we didn’t screen it for urase inhibitory activity in ref 47, while 5c,g are new compounds and fully characterized and screened for their urase inhibitory activity.

    R, R¢ were illustrated in Scheme one 1-4 a-c.

    we remove key intermediates and use derivatives, I completely agree with you, it is not just intermediates it is the main biologically potent urase inhibitors.

    It is very strange to see a description of NMR spectra (including two-dimensional ones) without pictures of the spectra themselves. Pictures of spectra, especially two-dimensional ones (on the basis of which the signal assignments were made) should be given at least in the Supporting Information, which is not available at all. This is more illustrative and gives the reader much more information than, for example, Table 1 (which can be deleted), especially since the assignment of signals in the spectra is presented in the experimental part. Table 1 shows the correlation between hydrogens and hydrogen-carbon which is missed in experimental part.

    Full spectroscopic details for compounds 5g and 7b have been included in the supporting information.

    I would recommend using the more commonly used names for compounds 6a and 6b: dimethyl and diethyl acetylene dicarboxylate, respectively.

    • Line 170 – Reference 49 is absent.

    Ref 49 is present, but the table position was not inserted in the right way so now it is OK.

    The caption to Figure 3 is unacceptable - here you should indicate what IC50 is and give the numbers of the compounds in the figure itself.

    We add the number of compounds in the figure.

    • Line 197 – Table 4?? Probably it should be Table 2.

    Yes we correct table number.

    • Figure 4 and Table 2 duplicate each other and one of them (figure) can be deleted.

    We deleted figure of the diagram and keep only table 2

    • Section 2.4 (Structure-Activity Relationship) is very short and needs to be expanded. Did the authors consider the possibility of synthesizing substances with two methyl groups in positions 1 and 6 of the quinolone fragment?

    It is not available to us to get 1,6-dimethyl quinolone.

    • In Experimental, Section 3.1 (Material and Methods) is empty. Sections 3.4 and 3.5 should be combined with Section 3.3 (Starting Compounds). The source of 6a and 6b should be indicated here as well.

    We merged 3.3, 3.4, 3.5 and 3.6.

    We completed starting material section and illustrate the source of 6a,b.

Reviewer 3 Report

Reviewers’ comments for the Manuscript ID: Molecules-1942839

The manuscript title:Synthesis and identification of new N, N-disubstituted thiourea, and thiazolidinone scaffolds bearing a quinolone moiety as urease inhibitors”. In the current manuscripts authors reports the design and synthesis of two series of molecules (5a-g and 7a-I) bearing N,N-disubstituted thioureas, and thiazole functionalities respectively  by modification of their previously reported molecules.  in vitro urease inhibition properties of newly synthesized molecules seem to be promising but design of further SAR needed. current manuscript may be suitable for publication in “Molecules journal after addressing below comments.

General comments

1)      Please rewrite the sentence from line 123-125, it is confusing

2)      Merge the scheme 1 caption in line 102 with reagents and conditions in line 99

3)      Correct the table-1, 13C-NMR values are overlapped

4)      It would be better to write compound numbers in figure 3

5)      Supporting information is missing, provide NMR spectra for all newly synthesized molecules

6)      In the conclusion part “The two series from quinolone ring bearing N,N-substituted thiourea, or fully substituted thiazolidinone rings  were designed with modifications at different positions. The N,N-disubstituted thiourea  scaffold with N-methyl quinolone system was the best” it is incomplete sentence. Best in terms what?

Author Response

Please rewrite the sentence from line 123-125, it is confusing

We did

2)      Merge the scheme 1 caption in line 102 with reagents and conditions in line 99

We did

3)      Correct the table-1, 13C-NMR values are overlapped

We corrected

4)      It would be better to write compound numbers in figure 3

We did

5)      Supporting information is missing, provide NMR spectra for all newly synthesized molecules

We did

6)      In the conclusion part “The two series from quinolone ring bearing N,N-substituted thiourea, or fully substituted thiazolidinone rings  were designed with modifications at different positions. The N,N-disubstituted thiourea  scaffold with N-methyl quinolone system was the best” it is incomplete sentence. Best in terms what?

We completed.

Round 2

Reviewer 1 Report

The article describes the results that provide useful information for chemists working in this area of research. The article has been revised taking into account the comments and can be accepted in present form.

Author Response

The article describes the results that provide useful information for chemists working in this area of research. The article has been revised taking into account the comments and can be accepted in present form.

Many many thanks

Reviewer 2 Report

The authors have made some corrections to the manuscript, but in this form it does not meet the standards of the journal.

I am very unpleasantly surprised by the absence in Supporting Information of pictures of NMR spectra of most of the new compounds (for 14 new compounds, the spectra of only two of them are given). This is completely inconsistent with modern standards for the characterization of new compounds and is unacceptable for journals of such a high level as Molecules. In addition, comparison of the available pictures of NMR spectra with their description in the text of the Experimental Part shows that the description of many signals does not correspond to the real picture of the spectrum (for example, instead of triplets, doublets of doublets with the same splitting constants are given).

I repeat once again that the presence of a well-processed spectral pattern is much more conducive to understanding than a hard-to-perceive table cluttered with numbers.

In response to a remark about the presence in the introduction of a large number of references that are not directly related to this study (the introduction contains 45 references out of 51 for the entire manuscript), the authors respond "the introduction part about the importance and the biological activity of each moiety in the targeted compounds (quinolone, thiazolidinone and thiourea) so these moieties represent a broad spectrum biologically active moieties and the targeted compounds may be further studied for other biological activities in the near future". In this case, we would like to advise the authors to use these references for further research in the near future, and not in this manuscript.

The authors also gave me a brief answer about "agricultural applications" but made no effort to convey what they meant to the reader.

Also, the proposal to expand the important section 2.4. Structure-activity relationship, the size of which does not exceed 6 lines, did not find a response from the authors.

Author Response

Third revised MS by the third referee

I am very unpleasantly surprised by the absence in Supporting Information of pictures of NMR spectra of most of the new compounds (for 14 new compounds, the spectra of only two of them are given). This is completely inconsistent with modern standards for the characterization of new compounds and is unacceptable for journals of such a high level as Molecules. In addition, comparison of the available pictures of NMR spectra with their description in the text of the Experimental Part shows that the description of many signals does not correspond to the real picture of the spectrum (for example, instead of triplets, doublets of doublets with the same splitting constants are given).

I repeat once again that the presence of a well-processed spectral pattern is much more conducive to understanding than a hard-to-perceive table cluttered with numbers.

Answer: The suppl file was modified and extended.

In response to a remark about the presence in the introduction of a large number of references that are not directly related to this study (the introduction contains 45 references out of 51 for the entire manuscript), the authors respond "the introduction part about the importance and the biological activity of each moiety in the targeted compounds (quinolone, thiazolidinone and thiourea) so these moieties represent a broad spectrum biologically active moieties and the targeted compounds may be further studied for other biological activities in the near future". In this case, we would like to advise the authors to use these references for further research in the near future, and not in this manuscript.

Answer: I really appreciate the advice of the reviewer; however, the assigned compounds would prospect to have broad biologically active activities, including anti-urease activity. However, I reduce the number of these refs.

Refs 9-13 were reduced to be only one ref [10]

The sentence Furthermore, thiosemicarbazides and their derivatives display various pharmacological effects, including antimicrobial, anticancer, anti-oxidant, anti-HIV, anti-viral, anti-sclerotic, anti-sclerotic, insecticidal, and anti-parasitic activities was removed

I removed the paragraph of…… Furthermore, thiosemicarbazides and their derivatives display various pharmacological effects, including antimicrobial, anticancer, anti-oxidant, anti-HIV, anti-viral, anti-sclerotic, anti-sclerotic, insecticidal, and anti-parasitic activities [9, 10]

I removed two refs from the biology of quinolone …..

In General, the number of refs were reduced.

The authors also gave me a brief answer about "agricultural applications" but made no effort to convey what they meant to the reader.

I removed that and I added a new ref [9] related to the pathological conditions

Also, the proposal to expand the important section 2.4. Structure-activity relationship, the size of which does not exceed 6 lines, did not find a response from the authors.

The SAR was extensively explained…..please look at That is shown as in case of compound 5c…….

Reviewer 3 Report

Reviewers’ comments for the Manuscript ID: Molecules--1942839

The manuscript title:Synthesis and identification of new N, N-disubstituted thiourea, and thiazolidinone scaffolds bearing a quinolone moiety as urease inhibitors”.

 Nicely, revised with all supporting information, so now the manuscript should be suitable for publication in “Molecules”.

Author Response

Comments and Suggestions for Authors

Reviewers’ comments for the Manuscript ID: Molecules--1942839

The manuscript title: “Synthesis and identification of new N, N-disubstituted thiourea, and thiazolidinone scaffolds bearing a quinolone moiety as urease inhibitors”.

 Nicely, revised with all supporting information, so now the manuscript should be suitable for publication in “Molecules”.

Many many thanks